# Exploring the Role of Champions in the Facilitation and Implementation of a Whole-School Health Program

Thomas Skovgaard *, Marie Louise Stjerne Madsen and Lars Breum Christiansen

Department of Sports Science and Clinical Biomechanics, University of Southern Denmark, 5230 Odense, Denmark; lsknudsen@health.sdu.dk (M.L.S.M.); lbchristiansen@health.sdu.dk (L.B.C.)
* Correspondence: tskovgaard@health.sdu.dk

**Abstract:** The aim of this article is to explore the role of local school champions in the facilitation and implementation of "The School Health Program". This study is based on semi-structured interviews with 26 local school champions. The interviews focused on exploring key implementation responsibilities and barriers and facilitators to implement core program components. Champions identify coordination, promotion, and handling of support functions as key implementation tasks. The findings highlight organizational and structural factors that impact this type of implementation agent. Teachers functioning as champions can be central agents in the implementation of a whole-school program. Champions must, however, be supported by thorough organizational preparation, engaged leadership, and a well-defined distribution of roles.

**Keywords:** school-based health program; champions; physical activity; implementation; training





## 1. Introduction

Schools are recognized as a key environment for promoting health and well-being among children and young people. Children and young people from across the population are in school for many hours, and schools supply important infrastructures for physical activity [1–5].

Schools are, however, complex and busy places where promoting health is just one of many aims that, it is argued, must be balanced in relation to the core goals of primary education, which in general, can be formulated as providing children with the knowledge, skills, and attitudes deemed important to thrive academically and personally. In many educational systems, this entails a strong focus on developing skills like literacy and numeracy together with transferrable capabilities related to critical thinking, problem solving and effective communication [6–9].

Implementing health initiatives in such a context requires considering, e.g., the functions, motivation, and qualities of innovation deliverers like teachers and school management; the particular environments where the innovation is put into practice; and the wider setting encapsulating the particular environments, such as school districts or municipalities [10]. A number of recent studies, in particular, emphasize the need for local implementation champions who can facilitate and support school-based health initiatives [1,9,11–14]. Teachers are well suited for this role due to their expertise and position within the school [15,16]. However, challenges such as busy schedules, a lack of support, and the need for additional training can hinder their effectiveness [1,17,18]. For reasons such as these, we need to know more about the specific operationalization and effects of the champion's role as a particular type of implementation agent in a school setting. For instance, it is yet to be explored, in full, what might help and support local champions when taking on new responsibilities and implementation tasks such as facilitating, motivating, and activating staff involvement in the implementation of a whole-school program [9,13,17,19,20].

For such reasons, the purpose of this article is to enhance our understanding of the role of local champions in the implementation of a specific whole-school program designed to promote health, physical activity, and learning among students in lower and secondary education in a Danish municipality. In this article, we aim to identify key implementation tasks and responsibilities for local champions and explore factors of particular importance for the implementation of core program components.

*The School Health Program*

The School Health Program (SHP) is a municipality-wide initiative in Denmark that ran from 2016 to 2021, involving 20 public schools. The primary objective of the SHP was to enhance learning by prioritizing health, well-being, and physical activity, resulting in a more active and varied school day. The program consisted of five core components: (a) establishing a local SHP committee at each school; (b) appointing 1–2 local SHP champions at each school, often teachers, including physical education (PE) teachers; (c) participating in specialized training courses to support quality PE and increased physical activity opportunities throughout the school day; (d) providing schools with materials and equipment; (e) and offering an open digital platform with resources and inspiration for local champions and school staff.

The first step in the SHP program was to establish a local committee and identify champions within each school. In addition to the champion(s), members of the local committee included a school management representative, a health visitor, and/or a school specialist with specific expertise of students' health and well-being. As part of the program, schools should develop a local plan of action and implement at least four local initiatives annually. These initiatives included classroom-based physical activities, various recess activities, initiatives focused on student well-being, and health-promoting activities such as encouraging healthy eating habits or promoting adequate sleep.

To support the implementation of local initiatives, all school staff, including school managers, general teachers, PE teachers, and SHP champions, participated in training courses during the program's initial years. The champions underwent a five-day training course, designed to equip them with the necessary skills to fulfill their role as implementation agents. The training course focused on advocacy, promotion, and communication, as well as basic knowledge of implementation processes and facilitating change within a school environment.

## 2. Materials and Methods

*2.1. Research Design*

A qualitative approach was chosen to explore the role and function of local SHP champions.

We conducted semi-structured interviews, a well-suited approach for gathering comprehensive insights into participants' perspectives, experiences, and viewpoints [21]. Additionally, we incorporated program-related documents as data sources to gain a deeper understanding of the primary objectives and prerequisites of the SHP, particularly as they pertain to local champions. The documents also provided formalized information on the purpose and specific content of the training course for local coordinators.

*2.2. Recruitment and Participants*

In total, 32 SHP local school champions across 20 different public schools (from small country schools to large city schools) located within one of the 98 Danish municipalities were invited to participate in the interviews. Out of all 32 champions, 26 agreed to participate in an interview.

*2.3. Interviews*

A total of 16 interviews were conducted. Ten interviews were held individually. Six interviews were carried out as group interviews; five with two participants and one

interview was held with six participants from a school for children with special educational needs. Due to the COVID-19 pandemic, a large portion of the interviews were held either as telephone interviews or online interviews (via Zoom). The interviews were conducted from September to November in 2020 and lasted between 20 and 40 min. Six local champions declined to participate primarily due to a demanding workload.

### 2.4. Interview Guide

A semi-structured interview guide, which provided an overall outline of topics including relevant follow up questions, was applied (see Table 1 for further details on interview topics and questions). Open-ended questions were used to gain insights, opinions, and experiences from local champions. The primary focus was their assessments of the SHP as a program, as well as an insight into their role and responsibilities as champion (e.g., key implementation tasks). Also, champions were asked to identify and elaborate on key barriers and facilitators, and how these influenced their work and function as implementation agents. Furthermore, to get an idea of local levels of implementation of core program components, we asked champions to provide a general overview of the process, priorities and results at their school. Finally, champions, based on their experiences from the SHP, were asked to provide suggestions for future school-based programs.

**Table 1.** Main themes: Interviews with champions.

| Theme | Examples of Questions |
| --- | --- |
| Descriptions and perceptions of the structure, content, and aims of The School Health Program | What parts of the program have you worked on, handled and developed the most? <br> How did you locally approach and operationalize the program? |
| Descriptions of and experiences with the implementation process—including, e.g., main barriers, enablers, and adaptations—of The School Health Program | At your school, has the program been adapted along the way, if 'yes', how and in what ways? <br> What have been the main factors enabling or inhibiting program implementation? <br> How has school management supported program implementation and your role as local school champion? |
| Descriptions and assessments of and proficiency in executing the role of local school champion | To what degree has the champion role been defined locally? <br> Have you participated in training and/or other courses related to the program and/or champion role? |
| Overall quality assessment and evaluation of the School Health Program | What has, for you and/or in your opinion, been the best thing about the program? <br> The purpose of the program was to enhance learning and well-being among students through a more varied school day: has your school achieved that? |

The interview guide was developed by the lead researchers and underwent rigorous testing, evaluation, and revision to ensure its clarity, coherence, and relevance. The objective was to develop a guide prompting unbiased, open-ended questions, designed to elicit rich responses. These steps were integral to ensure the guide's effectiveness and suitability for meeting the stated research objectives.

### 2.5. Interview Format and Informed Consent

The interviewer committed to attentive, non-judgmental listening, maintaining an open-minded stance to encourage frank discourse. By establishing solid protocols for information collection, including consistent recording methods and storage techniques, the veracity of data was secured. Furthermore, the interviewer was well versed in various interview techniques and procedures to corroborate information record responses accurately.

Prior to the study, all participants were provided with a detailed information letter that described the data collection and handling processes and outlined participant rights

and procedures to secure anonymity. Participants were asked to give verbal consent to sound recording before the interviews.

### 2.6. Data Analysis

To ensure consistency in the data collection process, all interviews were conducted by the second author of this article. Prior to data analysis, the interviews were anonymized, transcribed verbatim, and checked by a second researcher from the research group for accuracy. All transcripts were uploaded to the NVivo software (v. 12) for data analysis. As a guide, the qualitative data analysis used thematic analysis with both a deductive and inductive approach [22]. As an initial step, the primary researcher read all transcripts while noting preliminary coding ideas. This was carried out across the entire data set. During this initial analysis of the interview data, topics from the interview guide were used as a preliminary deductive coding scheme. Theoretically, the analysis was inspired by Durlak and Dupre's ecological framework for understanding effective implementation which highlights the importance of considering multiple levels of influence when analyzing the employment of programs, interventions, or policies. The framework recognizes that successful implementation is not just dependent on individual factors, but also the wider context in which implementation takes place. More concretely, starting from five overall categories, Durlak and Dupre's framework identifies the many factors that may influence effective implementation [23]. These categories were helpful as a preliminary coding scheme in the early analytical process. For instance, based on the responses from local champions, the scheme helped to identify important implementation factors at school level. Subsequently, all preliminary coding ideas were discussed to check for accuracy against the entire data set. In order to maintain an open-minded perspective towards the data, the primary researcher and the research group independently re-read the entire dataset. Again, notes and coding ideas were compared and discussed, and in the end, six key themes emerged from the data.

### 3. Results

The results are presented in six sections with the headings: Key Implementation Tasks; Training Course; Implementation Procedure; Support from Colleagues; Support from Management; and Teamwork.

### 3.1. Key Implementation Tasks

Most of the champions expressed a positive attitude towards the SHP program and believed that the school could promote healthy behaviors among their students through the program. One SHP champion stated: "This program gives us the opportunity to grasp health behaviors more broadly, and as a school it gives us the opportunity to help and support children to become healthier and more active" (Interview 7). *Coordinate, Promote, and Support* were identified as the key implementation objectives. Coordination mostly concerned scheduling and facilitating SHP committee meetings as well as distributing inspiration and resources from the central project office to colleagues. Moreover, coordination involved prioritizing, planning, and implementing local initiatives. However, interviews with champions revealed that only a small number of the schools were able to accomplish the four required initiatives per year. Promotion functions, such as *presentation of program intentions* as well as *recruitment of school staff to take part in local initiatives*, were also identified as important objectives. Finally, continuous support of colleagues was identified as a key task. For most interview participants, this meant supporting and motivating colleagues to try out new materials as part of local initiatives. An example of a program initiative, established at several schools, is classroom-based physical activity. One champion noted: "My job as a champion is to find relevant classroom activities and then promote these for try out among my colleagues" (Interview 10).

### 3.2. Training Course

All champions participated in the five-day introductory training course. The overall aim of the course was to provide champions with basic knowledge and skills on how to coordinate, promote, and support change within a school environment. One champion explained that the introductory training course mostly helped to define the *role as a coordinator* and how to adjust their role to specific local circumstances: "For me the role and my job were defined on the first day of the course. And they [course leaders, ed.] particularly highlighted the coordinative part" (Interview 10). Most champions acknowledged the important role they had in the coordination of local SHP initiatives. At the same time, they lacked a clear definition of what being an SHP champion exactly involved: "How big is this role, what is expected of me and not only in terms of my time, but also what it means to be a champion. What is 'the job'?" (Interview 3). This indicates an only partially clear definition of the *champions particular tasks and responsibilities* in the SHP. According to one interviewee, there is a discrepancy between the content and goals of the competency development program (i.e., the installed training course) and the school setting: "The course trained us in the coordinative skills (. . .) necessary for filling out the role as champions (. . .) However, our very first local committee meeting, made it clear that this coordinative function was not fully compatible with our context" (Interview 12). Some champions also mentioned that, for instance, coordinative and support tasks were not foreign to them as professionals: "I wanted to be our local champion (. . .) because I already coordinate activities related to physical education and physical activity in our school. It was natural for me to (. . .) become involved in this program" (Interview 2).

### 3.3. Implementation Procedures

To support local ownership and the implementation of core program components and initiatives, each school was required to establish a SHP committee and develop an action plan with at least four health initiatives each year. In fact, most schools did define program initiatives and objectives in an action plan. However, interviews with local champions made it clear that these plans were realized only to a small degree. The developed plans were not used actively to guide program delivery, for instance, by monitoring the quality of deliverables. Furthermore, the champions explained that it was necessary to adjust and modify program components to local context more thoroughly. In fact, several champions noted that without active modifications of program components, such as, for instance, committee meetings and local initiatives, local program implementation would have been complicated further.

Some schools succeeded to establish joint understandings of the entire program and its core components. The champions on these, often smaller, schools explained that they experienced a sort of *collective responsibility*. This was described as staff and school management, together, contributing to both the coordination and execution of local program initiatives. One champion pointed out that this sort of involvement, typically, was initiated at school meetings: "We used staff meetings to involve colleagues. Both in terms of selecting our local health initiatives and in carving out actual implementation plans. Our job as champions was to coordinate the workload" (Interview 3). *Collaboration and shared responsibility* among local champions, school staff, and management were key factors to enhance program implementation and local ownership. This sort of collectiveness occurred more regularly at smaller schools, where, with a more head-on approach, it was possible to involve all parties and easier to adjust program components to fit the local context.

### 3.4. Support from Colleagues

Most champions saw themselves as having an important role in promoting and supporting the implementation of the SHP program. However, they struggled with the position as an *informal leader mediator* between colleagues and school management. Several champions found it difficult to navigate this type of organizational cross field. The interviews made it clear that a problem at school level was a lack of general acceptance of the champion

role. The champions especially problematized the difficulty with recruiting, activating, and motivating colleagues to engage in and find the time for program initiatives. Overall, many interviewees questioned whether it was, in fact, their responsibility to encourage and recruit colleagues to take part in local program initiatives: "I find it difficult to recruit my colleagues to participate in something 'extra'. Often, I end up taking the full responsibility myself" (Interview 2). Most champions also reported limited support among school staff, especially when asked to find the time and energy to commit to the SHP program: "Often when I mention the program to my colleagues, they instantly say, 'What is this, why must I do it, and I don't have the time'" (Interview 1). This sort of *resistance and lack of prioritization* from colleagues was a constant issue for the champions: "My colleagues often question programs like this. They immediately ask if they have do it (...) And that question is probably, in itself, the biggest barrier" (Interview 1).

### 3.5. Support from Management

School management played an important part in the specification and prioritization of champion tasks and responsibilities. Moreover, interviews made it clear that active and encouraging management support was a key factor for not only establishing the local committee and implementing specific activities, but also for supporting the champions' legitimacy and place within the school organization. An active management was also important for the level of local ownership and buy-in from school staff in general. Positive and continuous reinforcement from school management, underlining the importance of the school's participation in the program, helped champions meet program intentions. As some of the champions explained, an engaged school management could make a difference in whether school staff accept the program or not. This type of *active leadership* was experienced at ten out of twenty of the schools. Champions at the remaining half of the schools identified school management as less active in their leadership role. Even though some support was provided, SHP champions reported that school management, in their opinion, especially lacked long-term interest and motivation in relation to the program. Champions also found it difficult to engage their school manager in local committee meetings: "Our school manager has not been a driving force in this program. For me it's difficult to be the only one pushing things forward" (Interview 13).

### 3.6. Teamwork

Close collaboration with a second champion was also identified as a key factor. Through this collaboration, co-champions were able to guide and assist each other and it was emphasized that it helped them to better accomplish coordination tasks, deadlines, and program responsibilities. Overall, *collaboration between two or, in one case, more champions* was identified as a strength of the local organization and implementation of program initiatives. One SHP champion highlighted that having a partner also enhanced the ability to motivate school staff: "Being two coordinators have absolutely helped us. We can inspire and help each other, and when introducing this program to our colleagues ... it's just easier being two" (Interview 3).

## 4. Discussion

The purpose of this study was to identify tasks and responsibilities for local champions in a whole-school health program and examine the key factors affecting their involvement in the SHP. The results underscore organizational and structural aspects impacting the work and legitimacy of local champions. These implementation agents are vital for efficiently executing tasks like coordination, promotion, and establishing enduring local support. They also bridge the gap between the overall program level and school-based activities and deliverables. Notably, champions often encounter challenges in their roles, particularly when lacking active engagement and commitment from school management and colleagues. A recurring theme across the interviews was the champions' need for legitimacy, shared accountability, and support from school management. This study adds to

previous research regarding the central role school managers have in the implementation of school-based programs [15,24–26]. Furthermore, this study supports the recommendation that school managers must work to secure an enabling implementation environment for local champions that makes it possible for them to perform effectively in their (new) role. This involves more than merely recognizing and formally authorizing their positions. For example, it is about encouraging other staff members to give priority to the program and support their champion. This perspective is supported by Carson and colleagues [9,27]. In general, the literature shows that leaders and leadership support implementation by guiding and empowering key implementation agents. Effective leaders also provide managerial backing, address challenges during change, and ensure that measures being implemented are in line with the organization's primary goals. Leadership's role in establishing organizational capacity for change and a clear and shared direction is fundamental for successful implementation. It is imperative for leaders to not only strategize and decide but also to motivate and foster creativity [28]. Similar considerations are stressed in the essential work by Durlak and Dupre cited earlier. In their analysis, based on a review of more than 500 studies of factors affecting implementation processes, establishing a shared vision, together with upholding strong stakeholder commitment and buy-in, is pointed out as a core organizational component impacting implementation.

Interestingly, working together on fulfilling the role as a champion is identified as a method to strengthen program implementation. Within implementation research, there is a growing interest in assessing the importance of coordinated and leadership-supported collaborations among key implementation actors. Often, this topic is addressed by focusing on the role of implementation teams as a means of increasing organizational capacity to implement, adapt, maintain, and scale, e.g., specific interventions or more general initiatives. The research on implementation teams is at a preliminary stage. This is even more the case for empirical studies testing the role and influence of such entities. However, an increasing body of knowledge points to teams as playing a crucial role, not least in more comprehensive implementation processes. This study supports such an inference.

Other factors, at an intercollegiate or individual level, may also come into play during the implementation process. Studies suggest that champions need solid transversal skills related to communication, advocacy, problem solving, and collaboration. These competencies help them in recruiting and training peers to participate in extended health-related programs [11,19,29]. However, qualifications such as these may go beyond what is expected of, for instance, physical education teachers. While many physical education teachers are familiar with components of health-related school schemes, further competence development and knowledge building related to specific program components is typically needed to take on the role of a champion [18]. In our study, the champions identified the preliminary training course as helpful and supportive, primarily in clarifying their coordinative role. The SHP champions requested more knowledge on, for instance, recruitment and promotion among colleagues; the general theory of change underpinning the program; and skills and insights on how to successfully coordinate implementation tasks and responsibilities. Webster and colleagues (2015) recommend that future champions be trained in performing organizational and administrative tasks; communicative and advocacy skills to help champions gain support among school staff; and strong promotion skills to successfully help the sustainability of successful school programs [13]. Again, these findings and recommendations corroborate earlier, key publications, like Durlak and Dupre, on factors influencing implementation. As a supplement to this, we suggest that future training opportunities should help the champion to initiate closer collaborations with school management and other members of school staff. Together, this may strengthen the effectiveness of local champions and, by extension, the success of school-based initiatives like SHP.

## 5. Conclusions

In our research, we discovered that champions can play a primary role in executing key tasks and responsibilities for the successful implementation of a whole-school program.

Nevertheless, the findings also emphasize that the responsibility for such multifaceted tasks cannot solely rest with the champion. We identified important factors within the school structure and organization that markedly influence the effectiveness of champions. Their success hinges on gaining legitimacy and establishing a positive working rapport with their colleagues. These key factors are facilitated by thorough organizational preparation, engaged leadership and solid management, and a well-defined distribution of roles. We suggest that one important area for future research on school-based programs, making use of champions in the implementation phase, is testing and establishing effective strategies to prepare and train champions to take on their role in ways that profoundly enhance program results and outcomes.

This study should be considered with certain limitations in mind: The SHP was a community-wide program involving all schools. Although the SHP shares similarities with other whole-school health programs, it is a distinct program implemented in a particular municipality in Denmark. Furthermore, though we were able to recruit and interview nearly all of the champions and gained in-depth insights into their practice and deliberations, it is possible that the program attracted a unique group of teachers with specific interests or qualifications for such tasks. These factors could potentially affect the generalizability of our findings.

*Implications for Future Practice*

The findings of this study can inform school management and champions in their collective work to improve student health. Three key factors to further optimize this are:

- Thorough and early program planning, focusing on all phases from initiation to potential long-term maintenance. Importantly, this includes mechanisms to ensure that key elements, proven to be particularly effective and/or important, are maintained post the primary implementation phase.
- School management as a driving force, providing visible and clear guidance and support to the program and its champions.
- A widely endorsed understanding of the champion's role, ensuring their actions and responsibilities are legitimized. This involves the methods, timing, and reasons champions employ to onboard, assist, and motivate their peers in implementation activities.

**Author Contributions:** Conceptualization, T.S., M.L.S.M. & L.B.C.; methodology, T.S. & M.L.S.M.; formal analysis, M.L.S.M.; investigation, M.L.S.M.; writing—original draft preparation, M.L.S.M.; writing—review and editing, T.S. & L.B.C.; project administration, L.B.C.; funding acquisition, T.S. & L.B.C. All authors have read and agreed to the published version of the manuscript.

**Funding:** The School Health Program was supported by the A.P. Møller Foundation (Grant No. 15-05-0031), as part of a large initiative to strengthen the Danish Folkeskole system, encompassing both primary and lower secondary education. This particular municipal effort was granted partial funding for the reported research study.

**Institutional Review Board Statement:** In accordance with Danish law (LBK no. 1338 of 1/9/2020; http://www.retsinformation.dk/eli/lta/2020/1338) no formal ethical approvement was required, because the study did not collect human biological material. The study and its data management procedures were approved (review number 10.205) by the Research and Innovation Organization (RIO) of the University of Southern Denmark. The study was conducted in accordance with local legislation and institutional requirements.

**Informed Consent Statement:** All participants were informed about the study and their participation was voluntary, and they could withdraw at any time.

**Data Availability Statement:** The data presented in this study are available on request from the corresponding author.

**Conflicts of Interest:** The authors declare no conflict of interest.

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
