# Peer review of "Exploring the Role of Champions in the Facilitation and Implementation of a Whole-School Health Program"

_education, doi:10.3390/educsci14020161_

Round 1
Reviewer 1 Report
Comments and Suggestions for Authors
Exploring the Role of Champions in the Facilitation and Implementation of a Whole-of-School Health Program. A Qualitative Study
I would like to thank the editor and authors for the opportunity to review this manuscript.
The topic of the study mainly focuses on the exploration the role of local school champions in the facilitation and implementation of “The School Health Program”.
The study is relevant to the current times and the findings demonstrate the role of “School-based Health Program”. The writing style is clear and the content well organized. However, there are a number of issues that the authors need to take into consideration:
Title of the study
- while the authors created a title that is somewhat relevant to the content of the manuscript. I have a question about whether authors should include "qualitative research" in the title of their study. In general, "qualitative research" is a research methodology and is not recommended in the title.
Abstract
- It should be re-modified as appropriate with the comments below.
Materials and Methods
--Lines 90 to 94 - In the participants section, the authors combined both the descriptions about the participants and how they collected the survey. The latter is more appropriate to be included in a section about “procedures”. Therefore, I hope to re-organise the chapter "Procedures".
--The process of selecting participants was described in detail, but the timing was not. When did the interviews take place and for how long? Please provide specific dates.
--It should include something about the veracity or ethics of the data collected in the study.
--“ Interview guide” ......I think it would be much more readable if the basic interview questions were presented in a table. And “The interview guide was developed by the lead researchers and thoroughly tested, evaluated, and revised prior to use.”.....How was the lead researchers tested, evaluated, and revised? Please be specific.
--Lastly, “Participants were asked to give verbal consent to sound recording before the interviews.” ..... This is a very important part of the ethics of research. In a new chapter, the ethics of the research could be discussed and bundled with the veracity or corroboration of the data collected in the study. These are the same issues of "validity and reliability" that are usually addressed in "quantitative research".
Results
-- The authors mentioned in “introduction”...“ In this article we aim to 1) identify key implementation tasks and responsibilities for local champions; and 2) identify and explore key factors of importance for the implementation of core program components.”
-- What this sentence tells me is that I would expect the results to be presented in two separate parts, meaning results for 1) and 2) separately. However, the results of this study are presented as one, and I wonder why. If possible, I would like to see the results regarding 1) and 2) organised and presented separately.
References
--make sure to follow the journal’s reference guidelines.
Comments on the Quality of English LanguageMinor revision of the English language is required.
Author Response
Dear Reviewer,
Thank you for your comprehensive evaluation of our manuscript and the productive suggestions you provided. We value the time and effort you dedicated to enhancing the quality and clarity of our work.
In response to your suggestions:
- We have revised the title, aligning with the rationale behind your recommendation.
- The abstract has been updated to mirror the revisions throughout the manuscript.
- It is suggested to separate the description of participants and information on data collection via interviews into two sections – one being named 'procedures'. This point has been almost entirely accommodated by adding a section under the heading '2.3. Interviews'.
- Further information is requested about when the interviews took place. Details on this have been added. The same goes for additional information regarding the veracity or ethics of the data, which is expanded upon in a new section '2.5. Interview format and informed consent'.
- A table is requested that indicates topics and basic questions used in connection with the interviews. Such a table has been prepared and added (cf. Table I. Main themes: Interviews with Champions).
- It is, very reasonably, observed that the article's introduction may suggest that the reader is to expect results etc. presented in two separate parts. That is, however, not the intention. The article aims to integrate findings, analyses, and further deliberations of both perspectives within the same sections on results, discussion, and conclusions. This approach is designed to illustrate the interplay between different implementation elements and processes. To convey this integrated presentation style while simultaneously informing the reader about the two primary perspectives, the article's aim is compiled into one sentence, complemented by minor related adjustments throughout the text.
- Lastly, all references have been reviewed to adhere to the journal's guidelines.
We trust that these revisions address your concerns effectively and we look forward to any further comments.
Reviewer 2 Report
Comments and Suggestions for Authors
The manuscript represents an interesting contribution to the field of promoting healthy lifestyles. Research of this kind, which delves deeper into the implementation of programmes that involve the community through qualitative studies, should be highlighted.
A number of formal corrections are recommended:
- Line 69: prog'am's
- Line 126: Dupree and line 275 Dupre and line 401 Dupre (unify).
-Line 267: Carson and colleagues (9,27). I think it should be checked whether reference 9 is correct there.
- The references in general should be checked, as they are not unified (for example, there are journals with the full name and others with the abbreviated name; the doi appears in different formats, etc.). It is recommended to follow exactly the rules of the journal.
Author Response
Dear Reviewer,
Thank you for your assessment of our manuscript and the productive suggestions you provided. We value the time and effort you spent to enhance the quality of our work.
In response to your suggestions:
- The typo in line 69 has been corrected.
- The misspelling of the last name (Dupre) has been amended.
- The incorrect use of the Carson et al. reference has been adjusted.
- Lastly, all references have been reviewed to comply with the journal's guidelines.
We trust that these revisions effectively address your concerns, and we look forward to any further comments.